# Insulin Resistance/Hyperinsulinemia as an Independent Risk Factor That Has Been Overlooked for Too Long

**DOI:** 10.3390/biomedicines12071417

**Published:** 2024-06-26

**Authors:** Serafino Fazio, Flora Affuso, Arturo Cesaro, Loredana Tibullo, Valeria Fazio, Paolo Calabrò

**Affiliations:** 1Department of Internal Medicine, School of Medicine, Federico II University, Via Sergio Pansini 5, 80135 Naples, Italy; 2Independent Researcher, Viale Raffaello 74, 80129 Naples, Italy; flora.affuso70@gmail.com; 3Dipartimento di Medicina Sperimentale, Università degli Studi della Campania Luigi Vanvitelli, 80100 Naples, Italy; arturo.cesaro@unicampania.it; 4UOC Medicina Interna, Azienda Ospedaliera di Caserta, 81100 Caserta, Italy; loredana.tibullo@gmail.com (L.T.); valeria.fazio1986@gmail.com (V.F.)

**Keywords:** insulin resistance, hyperinsulinemia, risk factor, type 2 diabetes, cardiovascular diseases, cancers, cellular senescence, neurodegenerative diseases, Alzheimer’s disease

## Abstract

Unfortunately, cardiovascular diseases and cancers are still the leading causes of death in developed and developing countries despite the considerable progress made in the prevention and treatment of diseases. Maybe we missed something? Insulin resistance (IR) with associated hyperinsulinemia (Hypein) is a silent pandemic whose prevalence is continually growing in developed and developing countries, now exceeding 51% of the general population. IR/Hypein, despite the vast scientific literature supporting its adverse action on the development of type 2 diabetes, cardiovascular alterations, tumors, neurological disorders, and cellular senescence, is not yet considered an independent risk factor and, therefore, is not screened in the general population and adequately treated. There are now numerous substances, drugs, and natural substances that, in association with the correction of a wrong lifestyle, can help to reduce IR/Hypein. We are convinced that the time has come to implement a prevention plan against this critical risk factor. Therefore, this manuscript aims to highlight IR/Hypein as an independent risk factor for type 2 diabetes, cardiovascular diseases, cancers, cellular senescence, and neuropsychiatric disorders, supporting our conviction with the available scientific literature on the topic.

## 1. Introduction

Despite continuous progress in prevention and therapy, cardiovascular diseases and tumors remain among the leading causes of death [1]. Perhaps we have overlooked something. Insulin resistance (IR) with associated hyperinsulinemia (Hypein) represents a neglected risk factor responsible for a silent pandemic, which is producing a large number of hospitalizations and deaths [2]. Regardless of copious scientific literature demonstrating its adverse action in promoting the development of cardiovascular diseases, tumors, and neurological disorders, IR/Hypein is still not considered an independent risk factor and, therefore, it is not screened in the general population and not treated for preventive purposes [3,4]. lR/Hypein has been increasing worldwide, and due to people’s unhealthy lifestyles, its prevalence has reached 51% of the population in developed and developing countries, and it is continuing to grow. IR/Hypein can precede the onset of type 2 diabetes even for 15 years [5,6]. Thus, remaining unrecognized for many years, it produces significant and widespread damage, so much so that severe alterations are already present in patients with diabetes at the first diagnosis, particularly at a cardiovascular level. For this reason, these patients are subjected, by guidelines, to a much more severe prevention plan [7].

## 2. Insulin Resistance/Hyperinsulinemia

Insulin resistance is a condition in which the resulting metabolic action of insulin in the target tissues is inadequate for a given quantity of insulin. Therefore, to maintain blood sugar values within the normal range, the pancreas must secrete more significant amounts of insulin, resulting in hyperinsulinemia (increased circulating insulin levels), an essential characteristic of insulin resistance [6]. The mechanisms underlying IR, and those through which high levels of circulating insulin produce damage to the human organism, are very complex and need to be fully clarified. There are two isoforms of insulin receptor (IRec), IRec-A and IRec-B. The different physiological roles of the two receptor isoforms still need to be fully understood. This would result from their different binding affinity for insulin-like growth factor (IGF) receptors, particularly IGF-2. However, the roles of IRec-A in neonatal growth and development and of IRec-B in regulating metabolism are now well established. It should be underlined that the dysregulation of IRec-A/IRec-B functioning is associated with the development of IR, cellular aging, and the increase in the proliferative activity of normal and neoplastic tissues, resulting in harmful effects [8]. In the presence of the condition of insulin resistance, there is a functional postreceptorial defect, mainly at the level of the phosphoinositide-3 kinase (PI3K) pathway. This latter mediates insulin’s metabolic action and nitric oxide (NO) formation by the endothelium and the vascular smooth muscle cells.

On the other hand, the mitogen-activated protein kinase (MAPK) pathway is not particularly affected, so non-metabolic actions, including the stimulus on proliferation and the stimulation of endothelin-1 (ET-1) secretion, caused by the imbalance of the two pathways, are overstimulated by increased levels of circulating insulin [6,9]. The different behavior of the increased levels of circulating insulin on the two main post-receptorial pathways of insulin action produces, over time, severe alterations in the target tissues.

Although the gold standard for diagnosis of IR is the euglycemic–hyperinsulinemic clamp, it is not conceivable that this would be used for mass screening [10]. Still, many surrogate tests could be used for this purpose. Among these, the best known are the Homeostatic model evaluation index (HOMA-IR) [11], the triglyceride glucose index (TyG) [12], the ratio triglyceride/high-density lipoprotein cholesterol (Tg/HDLc) [13], the fasting insulin levels [14], the glucose to insulin ratio (GIR) [15], and the visceral adiposity index (VAI) [16] Table 1. Most of these surrogate indices of IR have demonstrated an excellent correlation with the clamp results and could be used for this purpose [11,12,13]. Furthermore, these three surrogate indices were also found to be independent predictors of cardiovascular events [17,18,19].

The three indices are calculated very simply: HOMA-IR = fasting blood glucose (mg/dL) × fasting insulinemia (mU/L)/405, normal values between 0.23 and 2.5; TyG = lgn [fasting triglycerides (mg/dL) × fasting blood glucose (mg/dL)]/2, normal values < 4.65; fasting triglyceride (mg/dL)/fasting HDLc (mg/dL) ratio, normal values < 2. 

The observation that the incidence of type 2 diabetes has been increasing in recent decades [20] has made the scientific community focus on implementing measures to counteract this epidemic. The strategy appears to be finding glycemic value alteration before frank diabetes appears.

A recent manuscript suggested that, to predict gestational diabetes, a blood glucose cutoff value of 110 mg/dL (2 h after lunch) could be used [21].

Even more recently, the International Diabetes Federation published a position statement document declaring that a plasma glucose value ≥155 mg/dL one hour after the glucose load should be considered highly predictive for detecting progression to type 2 diabetes, micro and macrovascular alterations, fatty liver associated with metabolic dysfunction, and increased mortality in subjects with risk factors. All this is to screen subjects who will evolve towards prediabetes and overt type 2 diabetes before irreversible organ damage occurs and to intervene with appropriate treatments for preventive purposes [22].

It is not the intent of this manuscript to describe in detail the intimate mechanisms underlying the development of damage in different districts of the human organism by Hypein. 

However, we will briefly highlight some of the main alterations that can be produced by Hypein associated with IR.

### 2.1. Insulin Resistance/Hyperinsulinemia and Cardiovascular System

The scientific literature indicates that Hypein is responsible for endothelial dysfunction (ED), hypercholesterolemia, concentric remodeling of the left ventricle, and systemic arterial hypertension [for the prevalence of endothelin-1 (ET-1) on nitric oxide secretion (NO), the activation of the sympathetic nervous system, and the anti-natriuretic effect] [6,9] (Figure 1). In particular, ED is determined primarily by an imbalance at the vascular level in the production of NO and ET-1, favoring the latter, and secondly by the stimulation of endothelial and vascular smooth cell growth, both of which, together with the formation of vascular cell adhesion molecules (VCAM-1), promote the development and progression of atherosclerosis [23,24].

There is a lot of scientific evidence demonstrating how IR/Hypein is also linked with the development of increased left ventricular mass with concentric remodeling, which is also found in the diabetic heart [25,26,27]. It is well known that increased left ventricular mass index and concentric left ventricular remodeling are strongly associated with adverse cardiovascular events, particularly heart failure with preserved ejection fraction (HFpEF) [28].

We have already described in detail the alterations that IR/Hypein can cause to the cardiovascular system, so we will not go into further detail [29,30]. IR/Hypein exists in 30-70% of patients with heart failure (HF) and may be a significant contributory cause of the development and worsening of heart failure with preserved ejection fraction (HFpEF), which is characterized by a concentric remodeling of the left ventricle with diastolic dysfunction, increased filling pressures, and pulmonary and systemic congestion [26]. IR/Hypein is also an essential factor in the development of diabetic cardiomyopathy [27,28].

### 2.2. IR/Hypein and Cellular Senescence and Cancer

In recent times, it has also been demonstrated that cellular senescence (CS) is the link between aging and many associated chronic disorders and that CS is increased in adult obesity, type 2 diabetes, and non-alcoholic fatty liver, regardless of aging. IR/Hypein promotes the development of CS in human cells, particularly in metabolic targets, such as adipose tissue, muscle, liver, and brain [31,32]. Interestingly, among the various scientific manuscripts published in recent years, a study carried out in vitro on human hepatocytes subjected to chronic Hypein and, in vivo, on knockout mice for insulin receptors in the liver (LIRKO), demonstrated a direct link between chronic Hypein and hepatocyte senescence, and that these adverse effects of chronic Hypein on hepatocyte senescence can be blocked by reducing insulin receptors of hepatocytes or by the administration of senolytic substances such as dasatinib and quercetin [33]. It is supposed that hyperinsulinemia may play a role in tumorigenesis [4] even if the intimate mechanisms have not been completely elucidated. Recent observations in cancer patients have shown how Hypein may be a determining factor in influencing the development of obesity, diabetes, and cancer [34]. Both the insulin A receptor and the IGF-2 receptor mediate their effects through common oncogenic signaling pathways such as Ras/MAPK and β-catenin, which explains at least in part their involvement in carcinogenesis and the possibility that a chronic increase in circulating insulin levels leads to an increased risk of cancer in insulin-resistant subjects [35].

Excessive circulating levels of insulin act as a growth factor by binding not only to their own receptors but also to insulin-like growth factor receptors, with negative consequences on the development of tumors [4,6,9].

Some studies have shown that patients with type 2 diabetes or metabolic syndrome have an increased risk of cancer and cancer-related death [36,37]. However, it has also been reported, in subjects without diabetes or obesity, that hyperinsulinemia itself leads to increased cancer mortality, so much so that the authors conclude that improving hyperinsulinemia could be an important approach to preventing cancer [38].

### 2.3. IR/Hypein and Cognitive Impairment, Dementia, and Depression

For many years, it was thought that the brain was insensitive to insulin and was, therefore, not part of the target tissues of its action. Only in the last 20 years has scientific evidence accumulated showing that insulin crosses the blood–brain barrier, penetrating the brain where, by binding to its specific receptors, it regulates some functions of the central nervous system such as feeding, cognitive behavior, and depression, as well as controlling other essential systemic functions such as the production of hepatic glucose, lipogenesis, and lipolysis, and the sympathetic response to hypoglycemia [39].

Although the exact mechanisms underlying the actions of IR/Hypein in the brain have not yet been fully understood, it has been shown that there is a strict relation between IR/Hypein and cognitive impairment, dementia, and depression. In particular, in the last decade, a fair amount of interesting scientific literature has been published that supports this thesis. In a study published about ten years ago and performed on 328 patients with type 2 diabetes, it was shown that patients with higher levels of circulating insulin had worse cognitive function than those without hyperinsulinemia. Based on their results, the authors concluded that IR/Hypein must be considered a significant risk factor for cognitive impairment, especially with regard to delayed impairment in the memory domains [40]. Another study published in 2018 was carried out in Korea on 422 subjects who were over 65 years old and had normal cognitive function at baseline. During a mean follow-up of 5.9 ± 0.1 years, it was observed that the Korean mini-mental status examination (K-MMSE) decreased significantly in the participants who had a higher HOMA-IR index. Thus, the authors concluded that during a mean 6-year follow-up of elderly Koreans with normal cognitive function at baseline, increased IR/Hypein was significantly correlated with decreased cognitive function over time [41]. 

Dementia is characterized by a complex series of neuropsychiatric symptoms, among which cognitive function progressively deteriorates due to a series of degenerative changes in brain tissue. There are over 55 million people in the world affected by dementia, and 10 million new cases appear every year, so it has become a somewhat burdensome and expensive public health problem. Dementia is classified based on the cause that generates it, and Alzheimer’s disease (AD) is the most common form, accounting for approximately 60–70% of cases. It is characterized by the presence of deposits of beta-amyloid and a tangle of hyperphosphorylated Tau protein neurofibrils among brain cells. AD is so closely related to diabetes that it is also known as type 3 diabetes. It is a recognized fact that a decrease in cerebral glucose metabolism occurs at least ten years before the onset of AD [42]. According to the results of a Whitehall II cohort study carried out in the United Kingdom on 5653 patients, memory declined 45% faster, and the ability to make rational judgments regressed 29% faster in patients with diabetes [43].

Furthermore, IR/Hypein has also been shown to be related to major depressive disorders. In fact, a cross-sectional study performed on 1732 participants aged 26–36 years showed that men and women with depressive disorders had significantly higher indices of IR [44]. So much so that, based on the results of a study by Stanford University, it has been declared that IR doubles the risk of developing major depressive disorders [45].

## 3. Concluding Remarks

IR/Hyperins, through the development of type 2 diabetes mellitus, neoplasms, cardiovascular damage, neuro-psychiatric disorders, and cellular senescence, cause a worsening of the quality of life, an increase in hospitalizations with a notable rise in healthcare spending, and an excess burden of deaths. (Figure 2). It would be beneficial to inform the patients suffering from IR of the risks of this condition in order to improve their lifestyle, especially regarding a healthy diet and adequate physical activity. However, this is often not followed consistently over time or may be insufficient, so it may be necessary to add some treatments to help reduce IR/Hypein.

Today, we have numerous substances for treatment that reduce IR/Hypein, including drugs (particularly, SGLT2 Is, GLP-1As, and Metformin) and natural remedies (such as Berberine, Silymarin, Quercetin, L-Arginine, etc.), which have demonstrated effectiveness in reducing IR/Hypein both in experimental in vitro and in vivo studies and in clinical trials.

To sum up, we are experiencing a severe delay on this topic. In 2012, we published a manuscript in an attempt to stimulate the scientific community towards IR/Hypein, but without significant results [46]. We believe that it is mandatory to recognize IR/Hypein as a critical, independent risk factor for the development of cardiovascular and non-cardiovascular pathologies. A mass screening of the general population must be urgently planned in order to detect and treat IR/Hypein, with the aim of improving the quality of life of the population and attempting to reduce direct and indirect healthcare costs. Cardiovascular diseases, cancers, cognitive impairments, and dementia are key elements of global health in an aging society, so this represents a big challenge to provide adequate and sustainable care.

## Figures and Tables

**Figure 1 biomedicines-12-01417-f001:**
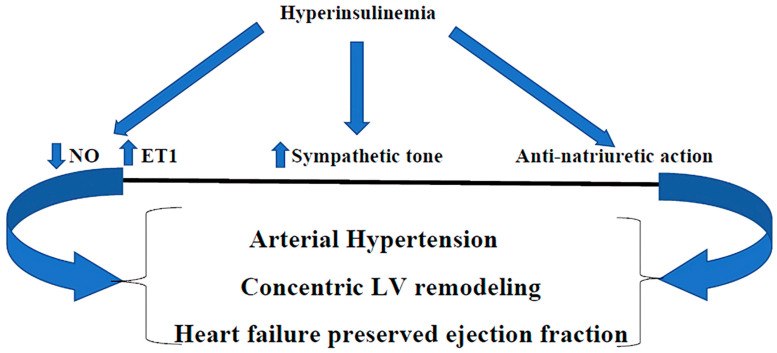
Hyperinsulinemia, through the imbalance between NO and ET1 (in favor of the latter), the stimulation of sympathetic tone, and the anti-natriuretic action, produces arterial hypertension, concentric LV remodeling, and heart failure with preserved ejection fraction. NO: nitric oxide; endothelin-1; LV: left ventricle. 
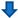
: Decreased; 
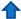
: Increased.

**Figure 2 biomedicines-12-01417-f002:**
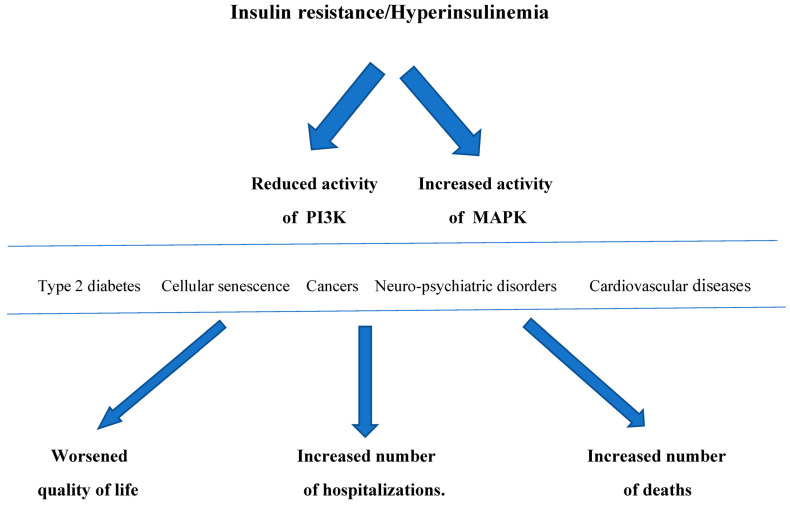
Insulin resistance/Hyperinsulinemia, through a reduction in PI3K activity and an increase in MAPK activity, determines type 2 diabetes, neoplasms, cardiovascular disease, neuropsychiatric disorders, and cellular senescence over time, which in turn result in worsening the quality of life, an increase in the number of hospitalizations with a notable increase in healthcare spending, and an increase in the number of deaths. PI3K: phosphoinositide 3-kinase; MAPK: mitogen-activated protein kinase.

**Table 1 biomedicines-12-01417-t001:** The table shows the most known surrogate indices of insulin resistance.

Indices	Metabolic State	Formula	Cutoff Values	References
HOMA-IR	Fasting	I _mU/mL_ × G _mg/dL_/405	>4.65	[11]
TyG	Fasting	Ln [TG _mg/dL_ × G _mg/dL_/2]	>9.36	[12]
TG/HDLc	Fasting	TG _mg/dL_/HDLc _mg/dL_	>2.75 for men and >1.65 for women	[13]
Insulin	Fasting	I _mU/L_	>12.2	[14]
GIR	Fasting	G _mmol/L_/I _mU/L_	<7	[15]
VAI	Fasting	Men: (waist _cm_/39.68 + [1.88 × BMI]) × (TG _mmol/L_/1.03) × (1.31/HDLc _mmol/L_); Women: (waist _cm_/36.58 + [1.89 × BMI]) × (TG _mmol/L_/0.81) × (1.52/HDLc _mmol/L_);	>0.34	[16]

Abbreviations: HOMA: homeostasis model of assessment; I: insulin; G: glucose; GIR: glucose to insulin ratio; VAI: visceral adiposity index; TG: triglycerides; HDLc: high-density lipoprotein cholesterol; TG/HDLc: triglyceride to high-density lipoprotein cholesterol ratio; TyG: triglycerides × fasting glucose.

## Data Availability

No new data was created or analyzed in this study.

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
