# Peer review of "Insulin Resistance/Hyperinsulinemia as an Independent Risk Factor That Has Been Overlooked for Too Long"

_biomedicines, 2024, doi:10.3390/biomedicines12071417_

Round 1

Reviewer 1 Report

Comments and Suggestions for Authors

In this opinion article the authors propose that insulin resistance with associated hyperinsulinemia should be considered as an independent risk factor for different pathologies. This manuscript provides essential and basic information on the effects of insulin resistance/hyperinsulinemia on different pathologies. The manuscript has been written for a general audience and the readers are appropriately referred to important published papers in the field. Nevertheless, I found that the manuscript contains redundant information which has been included in a recent paper from the same authors (ref. 24, Front Cardiovas Med 2024, doi: 10.3389/fcvm.2024.1380506). In fact, sections 2. Insulin resistance/Hyperinsulinemia and 2.1 Insulin resistance/Hyperinsulinemia and cardiovascular system have analyzed in deep in the previous work (ref. 24, Front Cardiovas Med 2024, doi: 10.3389/fcvm.2024.1380506). Hence, I suggest eliminating this information or to write a general statement reviewing the insulin resistance with associated hyperinsulinemia and skip the section Insulin resistance/Hyperinsulinemia and cardiovascular system (2.1). Moreover, I believe that some references are still lacking.

Author Response

Reviewer. In this opinion article the authors propose that insulin resistance with associated hyperinsulinemia should be considered as an independent risk factor for different pathologies. This manuscript provides essential and basic information on the effects of insulin resistance/hyperinsulinemia on different pathologies. The manuscript has been written for a general audience and the readers are appropriately referred to important published papers in the field. Nevertheless, I found that the manuscript contains redundant information which has been included in a recent paper from the same authors (ref. 24, Front Cardiovas Med 2024, doi: 10.3389/fcvm.2024.1380506). In fact, sections 2. Insulin resistance/Hyperinsulinemia and 2.1 Insulin resistance/Hyperinsulinemia and cardiovascular system have analyzed in deep in the previous work (ref. 24, Front Cardiovas Med 2024, doi: 10.3389/fcvm.2024.1380506). Hence, I suggest eliminating this information or to write a general statement reviewing the insulin resistance with associated hyperinsulinemia and skip the section Insulin resistance/Hyperinsulinemia and cardiovascular system (2.1). Moreover, I believe that some references are still lacking.

Authors. We thank very much the reviewer for his important suggestions. Accordingly, we have reduced the redundant information in section 2.1, regarding the effects of insulin resistance/hyperinsulinemia on the cardiovascular system and have added the reference indicated by the Reviewer.

Reviewer 2 Report

Comments and Suggestions for Authors

An article (opinion) “Insulin resistence/Hyperinsulinemia (IR/Hypein), independent risk factor…” is a compact but rather informative review the aim of which is to elucidate the role of  IR/Hypein as a risk factor for development of a lot of human pathologies including type 2 diabetes, cardiovascular alterations, tumors, neurological disorders etc  Biomedical significance of the review is obvious. At the same time there are a lot of shortcomings the correction of which would improve the quality of the presented material.

Firstly, may be better to change the title to IR/Hypein as an independent risk factor…”. There are some items in the text that requires deciphering: line 77 HDL (high density lipoproteins), line 84 T2D (possibly type 2 diabetes) while LV on line 207 must be excluded because it is not mentioned on Fig.2. The statement “the roles of IRec-A in neonatal growth and development and of IRec-B in regulating metabolism are now well established” needs reference/s especially considering the fact that as claimed by the authors less is known about physiological differences between the receptors.

There are only 2 Figures for visualization of the material so they must introduce maximum information described in corresponding section. Comments to Fig.1. Along with the effect on blood pressure it must also present the effect on concentric left ventricular remodeling and heart failure (HFpEF). At the end of the legend  to Fig.1 the word hypertension is missed  (line 114). It is better to use up- and down- arrows instead of “+” and “-“to mark the increased production of ET1 and decreased NO evolution.

A good assistant for doctors would be presentation in Section 2 all the “surrogate” tests for identification of IR/Hypein pathologies as a Table with a columns like: Test, Conditions (eg 2 hours after eat), Normal values, Reference.

The last comment. As neoplasms are mentioned among other pathologies induced by IR/Hypein impairment in Concluding Remarks the authors should substantiate this statement in more details in subsection 2.2.

The article can be published after correction.

Author Response

We thank very much the Reviewer for his suggestions.

  1. Reviewer. Firstly, may be better to change the title to IR/Hypein as an independent risk factor…”. There are some items in the text that requires deciphering: line 77 HDL (high density lipoproteins), line 84 T2D (possibly type 2 diabetes) while LV on line 207 must be excluded because it is not mentioned on Fig.2. The statement “the roles of IRec-A in neonatal growth and development and of IRec-B in regulating metabolism are now well established” needs reference/s especially considering the fact that as claimed by the authors less is known about physiological differences between the receptors.

Authors. According to the Reviewer’s observations we have changed the title, and corrected HDL with high density lipoprotein at line 77, T2D with type 2 diabetes at line 84, while have deleted LV at line 207. Furthermore, we have added a reference concerning the A and B isoforms of insulin receptor.

  1. Reviewer.There are only 2 Figures for visualization of the material so they must introduce maximum information described in corresponding section. Comments to Fig.1. Along with the effect on blood pressure it must also present the effect on concentric left ventricular remodeling and heart failure (HFpEF). At the end of the legend to Fig.1 the word hypertension is missed  (line 114). It is better to use up- and down- arrows instead of “+” and “-“to mark the increased production of ET1 and decreased NO evolution.

Authors. We have modified the figure 1 according to the Reviewer’s suggestions.

  1. Reviewer. A good assistant for doctors would be presentation in Section 2 all the “surrogate” tests for identification of IR/Hypein pathologies as a Table with columns like: Test, Conditions (eg 2 hours after eat), Normal values, Reference.

Authors. According to the Reviewer’s suggestion we have added in the section 2 a table with most of the surrogate indices of insulin resistance.

  1. Reviewer. The last comment. As neoplasms are mentioned among other pathologies induced by IR/Hypein impairment in Concluding Remarks the authors should substantiate this statement in more details in subsection 2.2.

Authors. To substantiate the statement regarding the relationship between insulin resistance/hyperinsulinemia and cancer, we have added a little paragraph on this issue in the subsection 2.2.

Reviewer 3 Report

Comments and Suggestions for Authors

Insulin resistance (IR) with associated hyperinsulinemia (Hypein) is a silent pandemic whose prevalence is continually growing in developed and developing countries. The manuscript aims to highlight IR/Hypein as an independent risk factor for type 2 diabetes, cardiovascular diseases, cancers, cellular senescence, and neuropsychiatric disorders. The opinion is supported with available scientific literature on the topic.

The article deserves attention and can be published after some technical adjustments. There is a feeling of some haste and carelessness when writing the manuscript. The drawings look rough and too banal even for such a short format as Opinion. The caption under Figure 1 is clearly incomplete. However, these shortcomings are easily corrected.

Author Response

We thank very much the Reviewer for his suggestion.

  1. Reviewer. The article deserves attention and can be published after some technical adjustments. There is a feeling of some haste and carelessness when writing the manuscript. The drawings look rough and too banal even for such a short format as Opinion. The caption under Figure 1 is clearly incomplete. However, these shortcomings are easily corrected.

Authors. We agree with the Reviewer and, according to his suggestion, have modified the the figure 1 and its caption

Round 2

Reviewer 1 Report

Comments and Suggestions for Authors

None